# Knowledge Gaps on the Utilization of Fossil Shell Flour in Beef Production: A Review

**DOI:** 10.3390/ani14020333

**Published:** 2024-01-21

**Authors:** Zimkhitha Soji-Mbongo, Thando Conference Mpendulo

**Affiliations:** Department of Livestock and Pasture Science, University of Fort Hare, Alice 5700, South Africa; tmpendulo@ufh.ac.za

**Keywords:** beef quality, climate change, natural feed additives

## Abstract

**Simple Summary:**

With the increase in the number of people, there is also an increase in the demand for livestock products, including good-quality beef. However, beef production is faced with the challenges of climate change while it also contributes to climate change. Although several strategies have been in place to address this synergy, they have constraints that do not enable them to be used resourcefully and by all types of farmers. There is a need, therefore, for new strategies that will address this synergy while ensuring that beef production will be sustainable, and quality will be preserved and/or improved. Fossil shell flour is one such strategy because it is cheap, readily available, eco-friendly, and can be used by all types of farmers. However, it has not been explored enough in research, and as a result, there is little to no scientific evidence to support its efficiency in sustainable beef production. There is, therefore, a need to address this research gap as a step towards sustainable and eco-friendly beef production.

**Abstract:**

Population growth in many countries results in increased demand for livestock production and quality products. However, beef production represents a complex global sustainability challenge, including meeting the increasing demand and the need to respond to climate change and/or greenhouse gas emissions. Several feed resources and techniques have been used but have some constraints that limit their efficient utilization which include being product-specific, not universally applicable, and sometimes compromising the quality of meat. This evokes a need for novel techniques that will provide sustainable beef production and mitigate the carbon footprint of beef while not compromising beef quality. Fossil shell flour (FSF) is a natural additive with the potential to supplement traditional crops in beef cattle rations in response to this complex global challenge as it is cheap, readily available, and eco-friendly. However, it has not gained much attention from scientists, researchers, and farmers, and its use has not yet been adopted in most countries. This review seeks to identify knowledge or research gaps on the utilization of fossil shell flour in beef cattle production, with respect to climate change, carcass, and meat quality. Addressing these research gaps would be a step forward in developing sustainable and eco-friendly beef production.

## 1. Introduction

Population growth and the enrichment of many countries are increasing the demand for increased livestock production and quality products [1]. Worldwide meat production has increased from 317 million metric tons to approximately 350.5 million metric tons from 2016 to 2023 [2] and is forecasted to marginally increase to a further 364 million tons [3]. This was accompanied by an increase in demand for higher-quality products [4,5]. Beef and veal have the third largest production volume, behind poultry and pork and ahead of sheep [6]. However, beef production represents a complex global sustainability challenge, including the need to meet the increasing demand and the need to respond to climate change and/or environmental footprints. Of all agricultural products, beef production requires the most land and water, and its production contributes the highest amount of greenhouse gas (GHG) emissions. Thus, the consumption of beef continues to pose several threats to the environment. On the other hand, the ability to predict the sensory and nutritional properties of meat according to production factors has become a major objective of the supply chain [7]. This evokes a need for sustainable beef production with the exploration of novel feed resources that will provide mitigation strategies for environmental impacts while not compromising beef quality.

The feasibility of using alternative feeds for ruminants depends, among other things, on the feed value of novel feeds, animal production responses, and feed costs compared to conventional feeds [8]. Several feed resources which include crop residues, Agro-industrial by-products, and non-conventional feedstuffs are already in use, but some of the constraints limiting the efficient utilization of these feed resources include low nutritive content, their conservation is challenging, some have antinutritional components, production often seasonal, and processing may be difficult.

A novel or under-explored feed additive like fossil shell flour (FSF) has the potential to supplement traditional crops in beef cattle rations, responding to environmental footprints while providing sustainable production and not compromising product quality [9]. Fossil shell flour, commonly known as diatomaceous earth (DE) is the remains of microscopic single-celled plants (Phytoplankton) called diatoms found in oceans and lakes in many parts of the world [9]. These remains have long been mined from underwater beds or ancient dried lake bottoms for decades and have numerous industrial applications.

Diatomaceous earth is mined, milled, and processed into various types for several uses. There are two main types of diatomaceous earth: food grade and filter/non-food grade [9]. Unlike the filter grade, the food-grade DE is commonly used in agricultural and food industries since it is considered safe for both humans and animals [9,10]. Fossil shell flour is non-toxic, cheap, and readily available in huge quantities in many countries [9]. This makes affordability and availability two of the greatest advantages of FSF for its use by any farmer [9]. In livestock production, FSF has recently been modified as an additive for several uses. It has been used as a feed additive, growth promoter, mycotoxin binder, water purifier, vaccine adjuvant, in inert dust applications in stored-pest management, a pesticide, and a natural source of silicon and anthelmintic [9]. Although FSF is beginning to gain interest in livestock production, more so in sheep, broilers, and layers, the use of FSF in cattle production remains unexplored.

The objective of this review is to explore knowledge gaps on the utilization of FSF in beef production. The use of FSF is evaluated based on sustainable beef production, beef quality, and requirements to respond to climate change and/or carbon footprints.

## 2. Fossil Shell Flour as a Feed Additive

The quality, flavor, and composition of beef change with the composition of the cattle’s diet [11]. The plane of nutrition has been found to influence meat quality due to its regulatory effect on the biological processes in muscle and in fat deposition [12,13,14]. Likewise, the type of forage fed to cattle affects both the carcass and meat quality characteristics of beef. Thus, recently beef quality including its fatty acid composition has been the focus of interest of many customers and researchers [15]. In most tropical countries, livestock is mainly reared under extensive production systems, where they mostly depend on natural pastures for their nutrients [16]. This type of production has the disadvantage of the scarcity of forage during the dry season, resulting in animals consuming a greater quantity of low-quality forage [17] and less palatable species resulting in an approximately 50% reduction in live weight gained during the wet season [18]. Likewise, feedlot cattle are also affected by climate change through impacts on forage and crop-based feeds. As a result, several feed additives have been used to supplement poor-quality feeds for livestock. These include antibiotics, probiotics, prebiotics, enzymes, antioxidants, mycotoxin binders, organic acids, beta-agonists, hormones, defaunation agents, herbal feed additives, and essential oils, which are mostly chemical-based [19]. It has, however, been indicated that, because of their chemical and physical characteristics, some of these feed additives could decrease feed intake due to a decline in liking and appetite of the consumed feed [20,21]. Also, recently, the use of these chemical-based feed additives has created concerns about chemical residues in meat and other animal products [22,23]. There is also an alarming ecological risk that is increasing with the accumulation of veterinary antibiotic residue in animal manure [24].

Due to the possible risks of chemical-based feed additives, there has been a rising interest in natural growth promoters (NGPs). Production systems have their interest inclined toward various plants and plant extracts, enzymes, organic acids, and oils as possible NGPs that are eco-friendly [25]. However, one of the major constraints in using these NGPs is the time and cost involved in their harvesting [9]. Nonetheless, one NGP that could substitute chemical-based feed additives, boost feed intake, and be useful as a cost-effective, readily available, healthy, and eco-friendly feed additive is fossil shell flour. Fossil shell flour is a natural, organic, silicon-rich substance that occurs as a soft sedimentary rock made up of fossilized relics of diatoms. It has important physical and chemical characteristics enabling its use as a feed additive, with mineral constituents that include Copper (30 mg/kg), Sodium (923 mg/kg), Zinc (118 mg/kg), Iron (79.55 mg/kg), Boron, 23 mg/kg, Magnesium (69 mg/kg), Vanadium (438 mg/kg), Calcium (0.22%), Potassium (0.08%), Magnesium (0.11%), Sulfate (0.062%), and Aluminum (0.065%) [26,27,28]. Although there is little to no information on the nutritive value of FSF, its richness in trace elements such as Zn, S, Cu, and Fe qualifies it as a possible solution to there being low levels of these minerals, especially in semi-arid regions, resulting in low growth rates and poor quality of livestock. Moreover, since it supplies more than 14 trace elements and other elements that are usually not available in abundance in most field crops [28], it may be used to correct nutritional mineral imbalances in livestock. A review by Ikusika et al. [9] details the physical and chemical properties of FSF as well as its uses in the animal industry and other human activities; readers are, therefore, encouraged to refer to this article.

Table 1 further summarizes some studies that have been conducted to research using FSF as a feed additive. Although there was no significant impact of fossil shell flour on poultry, the studies in Table 1 indicate that fossil shell flour influences growth performance parameters, diet digestibility, feeding behavior, feed acceptability/preference, and body condition score, with each improving with increased inclusion levels of FSF up to 6% or 60 g FSF/kg in sheep. In the study by Adeyemo [27], the authors have attributed the efficacy of the broilers to convert nutrients from feed into body tissue to the fact that fossil shell flour inclusion in animal diets daily tends to keep the animals free of parasites (particularly, worms) and toxic chemicals so they can reap maximum benefits from the feed and water they consume. However, it is not clear which compound in FSF is directly related to this phenomenon. The authors also reported an imbalance between calcium and other minerals in the diet, although there was a concomitant increase in the phosphorus content up to 1.5% inclusion of fossil shell powder, after which the phosphorus level dropped. The study by Ikusika [29,30] attributed the improved feeding behavior and/or acceptability of feed by rams to the rich Sodium, Calcium, Potassium, and Magnesium contents in FSF which improves the taste and aroma of the diets. The studies in Table 1 have, therefore, attributed the different effects of FSF on different animal parameters to its mineral content; however, there is no clear indication of the specific contribution of each element and which compound led to a specific result.

## 3. The Potential of FSF in Enhancing Carcass and Meat Quality in Beef

Recently, global meat consumption has been increasing along with concerns about food quality [32]. To meet this demand, growth promoters like antibiotics have been used; however, their use has been limited and/or banned in many countries due to the development of bacterial resistance that has alarmed the livestock sector [33]. As a result, natural additives have been opted for as a solution, as they have shown great potential to replace antibiotics by enhancing animal performance without changing or improving the meat quality [34,35,36]. However, these natural additives contain several compounds [37] that can be absorbed in the gut without being degraded and losing their main properties in the rumen, thus their properties like antioxidant activity may be transferred to the animals’ meat [35]. Consequently, this may have negative effects on the nutritional and sensory properties of meat depending on the antioxidant types. This, therefore, evokes a need for a natural feed additive that will enhance animal performance without negatively affecting the quality of the final product, i.e., meat, but will instead maintain or improve the quality and its safety for human consumption. Fossil shell flour is one such alternative.

The physical and chemical properties of FSF enabled it to be acknowledged as a natural animal health and sustenance product. Previous research has indicated that the inclusion of FSF in animal diets did not adversely affect the lean mass percentage of animals [38], improved the average body weight gain of cockerels [27,28] and sheep [29,30], the growth rate of piglets [39], and the body condition scores of sheep [30]. Among these growth parameters influenced by FSF, the growth rate has an impact on carcass traits, muscle and fat deposition, and meat quality attributes and appears to be the main factor contributing to the chromatic qualities of beef [40]. Although meat color and quality are not well correlated [41], beef color is the most important attribute for consumers as they use it to gauge the quality of fresh meat at retail points. However, 15% of retail beef cuts fail to meet the expectations associated with bright cherry-red lean color [42,43]. Thus, since most additives have been used yet there is still persistence in beef color not meeting expectations associated with a bright cherry-red lean color, this leaves room for research that will explore novel feed additives. Thus, it is important to test the effect of FSF-enhanced growth rate on beef color to respond to this knowledge gap.

Furthermore, other growth-related attributes like body weight gains and body condition scores are positively related to some carcass and meat quality characteristics. In a study by Apple et al. [44], the body condition of culled beef cows at slaughter influenced carcass quality and cutability characteristics, and further had an impact on the subprimal cut yields. Moreover, research has also produced various results on the link between carcass weight and meat quality based on different experimental conditions [45], with some authors reporting advantages while others have reported the negative influence of heavier carcasses on meat quality [46]. Among other factors that are attributed to heavier carcasses, are growth steroidal enhancers [47,48]. Although the knowledge of animal responses towards current growth enhancers is common, recent reports on currently used natural feed additives have shown that they can enhance animal performance without changing or improving meat quality [34,35,36]. Thus, variability and inconsistency in bovine carcasses and meat are still high [7]. Recent reports have shown that fossil shell flour can also be used as a growth enhancer [9,49]; however, nutritional manipulation using FSF to enhance growth parameters that are positively related to some carcass and meat quality characteristics remains unexplored. Addressing this knowledge gap may also provide solutions to the inconsistency in bovine carcasses and meat quality.

## 4. The Potential of FSF in Mitigating the Negative Climate Effects of Beef Production

The increasing demand for livestock products including beef due to the increasing population requires an increase in livestock production. On the other hand, the potential impacts of climate change on current livestock systems, while livestock production is also a contributing factor to climate change worldwide, are a major concern. Therefore, the interaction between increasing livestock production and ongoing climate change makes it challenging to increase production while lowering climate impacts [50]. One of the largest negative effects of climate change on meat production is heat stress. Meat production is affected by heat stress for all major commercial livestock types [51]. To decrease metabolic heat production, animals tend to reduce feed intake as an adaptive response to chronic heat stress [52], which consequently has implications on carcass deposition, carcass yield, and intramuscular fat content [53]. Heat-stressed ruminants exhibit reduced body size, carcass weight, fat thickness, and lower meat quality [54,55,56].

There is a lot of research that has focused on mitigation and adaptation strategies for climate change, which include, among others, adequate shade and water provision, sprinklers, and air conditioners. However, these strategies are not universally applicable, some are limited by dietary needs for specific productions, and availability, and they are too costly, and/or resource-intensive to be afforded by all types of farmers. Hence there is a need for alternative management practices that can be universally applicable and reduce heat load without affecting animal performance. One such product is fossil shell flour, which is readily available, cheap, environmentally friendly, and can be used by all farmers.

Recent research by Mwanda [57] that investigated the effects of fossil shell flour supplementation on heat tolerance of Dohne–Merino rams showed that water and feed intake increase with increasing levels of FSF, while the physiological parameters (skin temperature, rectal temperature, respiratory rate, and pulse rate) declined as the levels of FSF increased. The authors concluded that fossil shell flour could be used as a supplement in Dohne–Merino rams’ diet to mitigate heat stress and promote the overall productivity of the sheep. Furthermore, a study by Kellaway and Colditz [58] indicated that Friesians responded to heat stress by decreasing N retention while nitrogen losses were evident through urinary excretions. A study by Ikusika et al. [59] also investigated the effect of varying inclusion levels of fossil shell flour on growth performance, water intake, digestibility, and N Retention in Dohne–Merino Wethers. The study showed that a 4% inclusion rate of FSF will give the best improvement in growth performance, diet digestibility, and N retention of Dohne–Merino sheep. The authors further indicated that the addition of FSF in the diets of sheep is a safe natural additive that can help to reduce environmental pollution by reducing fecal and urinary N excretion. Nitrogen is an essential nutrient critical for the productivity of ruminants, if it is, therefore, excreted in excess, it becomes an important environmental pollutant contributing to climate change. So, ruminant feed manipulation using FSF as an additive may increase N retention and alleviate environmental pollution caused by urinary N excretions [60]. Although conclusions cannot be drawn on the impacts of FSF on climate change-induced heat stress based on limited scientific evidence/data, feed manipulation using fossil shell flour to mitigate heat stress and alleviate climate change effects needs to be explored more for sustainable beef production.

## 5. Limitations of the Use of Fossil Shell Flour in Beef Cattle Production

i.Due to the large body size and large amount of feed needed in beef production, the use of FSF will be very challenging, as large amounts of FSF will be needed. This will have a great effect on mining areas due to the demand for fossils and the deleterious impact on the environment and climate change.ii.Although FSF is known to be completely safe and non-toxic, there may be toxicity that may be associated with the interaction between the mineral content of FSF and that of animal diets, especially when it comes to heavy metals. For instance, FSF is suggested to have a concentration of 79.55 mg/kg of iron, while the daily dietary requirement in cattle is 50 mg of iron per kg of feed, Thus, possible interactions between fossil shell flour mineral contents and basal mineral contents of animal diets remain a research gap that needs to be further explored.iii.The feasibility of using alternative feeds for ruminants depends among others on the nutritive value, so since the nutritive value of fossil shell flour is also unknown, it is often difficult to determine the impact of its constituents on certain results in many studies. This may result in challenges in mixing well-balanced ratios of the essential nutrients in livestock feeds.iv.There is a dearth of information or studies to validate the safety of fossil shell flour and recommend safer and/or optimum inclusion levels for specific production purposes and species. This may result in farmers being reluctant to use FSF, especially small-scale farmers.v.Although FSF inclusion of 4% is suggested to increase N retention and aid in environmental pollution control, nutritional measures using FSF to reduce N excretion affect enteric methane emissions. For instance, a study by Ikusika [61] indicated that 4% and 6% inclusion of FSF in sheep diets increased enteric methane emissions. Thus, at certain inclusion levels, FSF can harm the environment.

Figure 1 (see Figure 1 below) depicts a solution approach to the limitations.

## 6. Conclusions and Recommendations

The need to meet the increasing demand for quality beef while responding to climate change represents a complex global challenge that needs novel, universally applicable, and sustainable interventions. Several novel and underexplored feed additives like fossil shell flour have the potential to supplement traditional crops in beef cattle rations to respond to this complex challenge. Fossil shell flour has not gained much attention from scientists and farmers, particularly small-scale farmers, and its use has not yet been adopted in most countries. The few studies that have been conducted on fossil shell flour have focused on small stock with an emphasis on growth performance. There is a dearth of information on the potential of using fossil shell flour in beef production. Research is needed to identify the potential of FSF in improving beef production, enhancing carcass and meat quality, and mitigating climate change effects in the context of all types of farmers and different countries. Addressing these research gaps would be a step forward in developing sustainable beef production.

## Figures and Tables

**Figure 1 animals-14-00333-f001:**
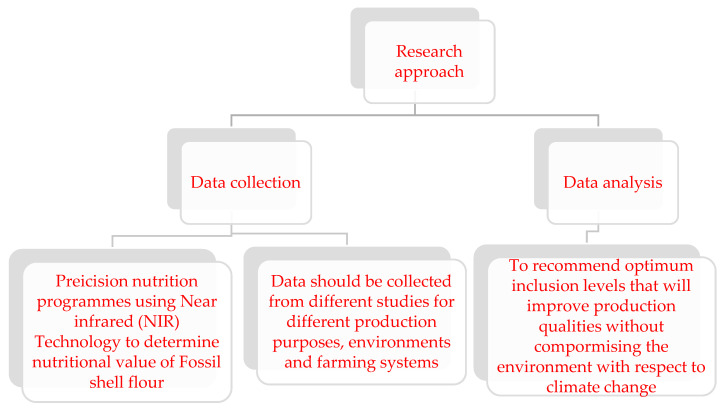
Limitations’ solution approach.

**Table 1 animals-14-00333-t001:** Some studies that were conducted on fossil shell flour.

Author and Year of Publication	Brief Methodology	Species	Country	Summary of Findings
Ikusika [29]	Twenty-four Dohne Merino rams were completely randomized and individually housed in pens for 90 days. Four different supplementation levels of FSF (0, 20, 40, and 60g/kg) were considered treatments for the rams.	Sheep	South Africa	The average daily feed intake, body condition score, average daily weight gain (g), and coefficient of preference (CoP was evaluated as the proportion of diet consumed by an individual to all the diets’ standard intake) were significantly higher in rams supplemented with 60 g FSF/kg than the other treatments. The order of preference of diets supplemented with FSF in feed intake by Dohne Merino rams was: 60 g FSF/kg > 40 g FSF/kg > 20 g FSF/kg > 0 g FSF/kg.
Ikusika [30]	Twenty-four wethers, weighing 20 ± 1.5 kg on average were fed dietary food-grade fossil shell flour in a completely randomized design of four treatments with six wethers in each treatment. The wethers were fed a basal diet without FSF addition (control, 0%), or with the addition of FSF (2%, 4%, or 6%) into the diet for 105 days.	Sheep	South Africa	Using fossil shell flour supplementation in the diets (2%, 4%, and 6%) improved dry matter intake, average daily weight gain, and body condition score as well as influenced feed preference and wool production and quality of Dohne–merino wethers.
Emeruwa [31]	Sixteen rams (18.5 ± 1.05 kg) were allotted to four treatments: Tl (0% FSF), T2 (2% FSF), T3 (4% FSF), and T4 (6% FSF) in a twelve-week growth study.	Sheep	Nigeria	The inclusion of 2.0% fossil shell flour in the diet of West-African dwarf sheep improved dry matter intake and reduced weight loss during lactation, while the inclusion of 4.0% enhanced the daily weight gain.
Adeyemo [27]	A total of 120-day-old broiler chicks were used for the experiment and randomly allotted to 5 treatments (T1—0.9%, T2—1.2%, T3—1.5%, T4—1.8%, and T5—0% inclusion levels, respectively).	Broiler chickens	Nigeria	Fossil shell inclusion had no significant influence on feed intake and feed conversion ratio but had a significant impact on weight gain. Values for feed intake and feed conversion ratio showed no significant differences (*p* > 0.05) among the treatment means. Results showed that for feed intake, there were no significant differences (*p* > 0.05) observed, and all treatments had the same mean value. For feed: gain ratio, T4 had the highest value (2.91) while T1 had the lowest value (2.31). Weight gain, however, showed significant differences (*p* < 0.05) between T1 and T4 (1.30 and 1.03, respectively). Results for the finisher phase showed no significant differences (*p* > 0.05) and were observed for weight gain, feed intake, and feed–gain ratio. Values for feed–gain ratio showed that T4 had the highest value (3.10) while the control (T5) had the lowest value (2.06). However, T2 (4.39) and T5 (3.62) had the highest and lowest values, respectively for feed intake. Weight gain values showed that T1 (1.78) and T4 (1.36) had the highest and lowest values, respectively.

## Data Availability

Data are contained within the article.

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
