# Peer review of "Knowledge Gaps on the Utilization of Fossil Shell Flour in Beef Production: A Review"

_animals, 2024, doi:10.3390/ani14020333_

Round 1
Reviewer 1 Report
Comments and Suggestions for Authors
Dear authors,
You find my comments below:
Line 19-33: half of the abstract content has general and introductory information about fossil shall flour (FSF). Concrete information about the FSF in summaries can be found in two sentences, lines 26-29, which is not enaught. The summary gives the impression from the start that there is a lot of discussion and debate besides the main topic, probably because there is not enough data for this topic.
Line 28: why did it not receive the attention of researchers if the authors claim that it has so much potential?
Line 31: "the authors believe" - this statement is not appropriate for a scientific article and especially in abstract section.
Line 37-59: only general informations, nothing focused on FSF. Too much extra aspects before entering the info about the FSF.
Line 74-78: the authors claim that FSF was used as a feed additive...and anthelmintic but no research is cited to support these claims in a review.
Line 83: diet composition instead of "diet regimen"
Line 83-109: only general aspects, no conctret information about FSF. A number of 26 lines with extra information and 20 lines with info focused on FSF. This approach shows that there are still not yet enough data to support the FSF topic into a review.
Line 82: in this section of "Fossil Shell flour as a feed additive" a chemical composition and nutritional value of FSF should have been presented from which its potential as a feed additive based on it's nutrients or compounds can be derived. In this regard, only lines 114-117 contain certain data. But the authors show the general way of using FSF in the feeding of some animal species. In its current form, this section is incomplete.
Line 131: Table 1: the "Title" column must be deleted, it is not necessary to put there because you cited the research by the author.
Table 1: what represents the "coefficient of preference"? Since this statement is included in the text, it must be supported by a formula.
The authors state that FSF has a significant effect on weight gain on broilers. Significant increase or decrease effect? At what age of days? Which feeding technological phase?
The authors present in table 1 the effect of FSF on the productive performances in sheep and broilers, but not show which compound/compounds of FSF led to these results, this would be of greater interest. Also, since the researches has been done on the productive effects, there are no information about carbon footprint reduction? But in the title and in the text they keep talking about this attribute of the FSF.
Line 134-160: only general information, nothing concrete about FSF. Very general approach.
Line 169-171: this information should be found in the previous section.
Line 174-180: Again here only additional information, the authors go into extra topics and definitions that should only be mentioned in a few words. By going through the text, the approach of the authors does not make you stay focused on FSF, but always disconnects and takes you to other topics.
Line 187-200: This extra information has nothing to do with the FSF subject. My impression is that here are some of the general factors that affect the meat quality. After this follows a phrase about the FSF (lines 201-202).
Line 202-205: it cannot be write that since FSF is a natural additive, and natural additives have been shown to enhance animal performance, therefore FSF could also do this, because each additive has a different structure and properties that gives its this wuality and from the data presented about FSF in this review, from my point of view, this hypothesis is not supported by data.
Line 207-226: general information about different topics that impact climate changes and greenhouse gas emissions and livestock impact.
Line 235-240: from a single research that addresses to several factors such as growth performance, water intake, digestibility, and N Retention, from my point of view the conclusion cannot be clearly drawn that FSF help to reduce environmental pollution. No clear data is presented to show the reduction of the carbon footprint on entire manuscript.
Reviewer 2 Report
Comments and Suggestions for Authors
The work is good, it involves the contribution of a very important topic socially speaking. I think I could expand the information a little more since the work is short for the relevance it represents. I suggest discussing the topic more broadly including more up-to-date bibliography.
Reviewer 3 Report
Comments and Suggestions for Authors
The topic of the manuscript is relevant and current. The manuscript is well written and easy to follow. However, I think the authors should consider including a paragraph where they include "limitations" of the use of fossil shell flour in beef cattle production, namely heavy metals, among others.
Reviewer 4 Report
Comments and Suggestions for Authors
Type of the Paper (Article
From the justification of the study on lines 67 to 77, it was indicated that FSF is used in livestock, but not in cattle, kindly mention the other livestock species it is used.
Critical statements were made in lines 123 to 128 and also 149 to 153, but were not referenced.
Due to the large body size and large amount of feed needed in beef production, the use of FSF will be very challenging, as large amounts of FSF will be needed. This will have a great effect on mining areas due to demand for the fossils and deleterious impact on the environment and climate change. These and other negative effects should be discussed.
What about the toxicity that may be associated with some of the minerals in FSF, especially for iron with a daily dietary requirement of 100 mg/kg, but a concentration of 79,55mg/kg (does this mean 7,955mg/kg or 79.55mg/kg) as indicated in the manuscript. Toxicity of the minerals in FSF versus normal dietary requirements should be discussed.
As indicated on line 239, how can natural additives (in this case FSF) help to reduce environmental pollution?
The review has little or negligible relationship with climate change and environmental footprints, thus this sentence “The need to meet the increasing demand for quality beef, while responding to climate change and/or environmental footprints represents a complex global challenge that needs novel, universally applicable, and sustainable interventions.” in the first part of the Conclusion section should be deleted.
Reviewer 5 Report
Comments and Suggestions for Authors
Thank the authors for submitting the manuscript titled “Knowledge gaps on utilization of fossil shell flour in beef cattle production, carbon footprints, carcass, and meat quality: A review”. This paper aims to identify knowledge or research gaps on the utilization of fossil shell flour in beef cattle production, carbon footprints, carcass, and meat quality. Although there are some novelties, I still have some comments for the authors to improve this manuscript.
1. could you please list some representative studies that can support your statement in Line 60-62, Page 2?
2. this is a review paper, but Table 1 only summarizes four studies on fossil shell flour. I suggest that the authors could consider adding more studies.
3. Similar to the first comment, could you please add references regarding ‘previous research’ mentioned in Line 167? Please also check other sentences that have the same issue like in Line 221.
4. in general, I think it would make this manuscript more appealing to be published if the authors could review more studies and make this manuscript a more extensive analysis.
Comments on the Quality of English LanguageThe quality of the English language is good for understanding.
Round 2
Reviewer 1 Report
Comments and Suggestions for Authors
Dear authors,
Thank you for yours responses point by point at each comment, but my decision remains to reject the publication of this manuscript, because from a nutritional point of view it is not correctly formulated and does not represent potential interest for animal nutrition as long as there are other economically and productively viable sources, easily accessible.
